# Proteostasis is adaptive: Balancing chaperone holdases against foldases

**Adam MR de Graff** [1] *, **David E. Mosedale** [1], **Tilly Sharp** [1], **Ken A. Dill** [2], **David J. Grainger** [1]

1 Methuselah Health UK Ltd, Cambridge, United Kingdom, 2 Laufer Center for Physical & Quantitative Biology, Stony Brook University, Stony Brook, New York, United States of America

* adam.degraff.work@gmail.com

## Abstract

Because a cell must adapt to different stresses and growth rates, its proteostasis system must too. How do cells detect and adjust proteome folding to different conditions? Here, we explore a biophysical cost-benefit principle, namely that the cell should keep its proteome as folded as possible at the minimum possible energy cost. This can be achieved by differential expression of chaperones–balancing foldases (which accelerate folding) against holdases (which act as parking spots). The model captures changes in the foldase-holdase ratio observed both *within* organisms during aging and *across* organisms of varying metabolic rates. This work describes a simple biophysical mechanism by which cellular proteostasis adapts to meet the needs of a changing growth environment.

**Data Availability Statement:** All relevant data are within the manuscript and its Supporting Information files.

**Funding:** The authors received no specific funding for this work.

## Author summary

Cells must maintain low levels of protein unfolding to avoid deleterious outcomes such as protein aggregation, oxidative damage, or premature degradation. The proteins responsible for this, called chaperones, come in two main varieties: ATP-consuming "foldases" that help clients fold and ATP-independent "holdases" that hold unfolded proteins until a foldase arrives. While foldases are necessary for folding, they are expensive to have in high quantities. Given that chaperones are abundant and costly, cells are under strong selective pressure to find economical combinations of foldases and holdases for maintaining low levels of unfolded protein. Yet, it is presently unclear what the ideal combination is and how it varies with growth conditions. By examining a toy model of chaperone function and minimizing the total cost of folding at different rates of protein synthesis, we find that while foldases are necessary at fast growth, holdases become increasingly effective at slow growth. Unexpectedly, total chaperone requirements were predicted to increase as synthesis slows, consistent with observations across age in worms, as well as across species with varying metabolic rates. This work thus provides a general framework for understanding the chaperone requirements of a proteome in terms of an energy minimization principle.

**Competing interests:** The authors have declared that no competing interests exist.

## Introduction

Proteostasis is a big activity of a healthy cell. It entails the protein synthesis, chaperoning and degradation needed to sustain a folded, functional proteome. It is costly, consuming as much as 25–50% of a cell's energy [1]. That energy goes into synthesis, and making and running chaperones [2–4]. Proteostasis needs to be adaptive, operating differently depending on the cell's growth rate, stress levels, or other environmental factors. Different external conditions require the differential synthesis, folding and chaperoning of different proteins. And different amounts of energy are available under different conditions. How does the cell determine its chaperone needs? And, what is the nature of its response?

We propose here a simple biophysical principle that can implement proteostasis adaptation. We start from the distinction between the two types of chaperones: *foldases*, which are energy-costly and ATP-dependent [2], and which accelerate the transition of non-native conformations towards native states [3], and *holdases*, which do not use ATP and which simply protect their client proteins from aggregation [5,6]. We hypothesize that the cell's fitness objective–i.e. what the cell "cares about"–is to keep its proteome as folded as possible, with the least expenditure of energy.

Previous work on holdases has focused on their ability to affect aggregation, primarily by keeping non-native protein soluble in a refolding-competent state [5,6]. These studies often apply high stress conditions that rarely occur in larger homeotherms like us. Yet, even under low stress laboratory conditions, holdase overexpression can extend lifespan [7,8] and under-expression shorten it [7,9]. The aim of this work is to use a simplified model of holdase-foldase function to explore mechanisms by which cells and proteomes may benefit from different ratios of these two functions under low stress, steady state conditions.

## Results

### The different actions of holdases vs. foldases

Proteostasis can be regarded as a complex traffic pattern on a network: different client proteins in different states of folding or misfolding, trafficked through the cell's different types of chaperones. Those details are modelled elsewhere [2,10]. Here, Fig 1 shows the simplified version that best serves our purposes here. First, we divide chaperones into just two classes, holdases and foldases. Second, we center our attention on the unfolded state $U$, not the native state $N$. The steady state concentration of free (non-bound) unfolded protein $[U_{free}]$ (shown here metaphorically like a liquid level in a container) is sustained by a balance between an inflow and outflow. The two inflows are of newly synthesized protein and from native protein that is unfolding due to finite stability [2,11,12]. The diagram shows two circuits: (a) an energy-dependent foldase circuit that gives its client a chance to fold and (b) an energy-independent holdase chaperone that captures $U_{\text{free}}$ and provides a "holding tank" for the client until a foldase arrives to fold it. Based on the known binding and rate coefficients (S1 Table), we can consider circuit (a) to be in steady state equilibrium on the time scale of action of circuit (b).

First, consider (a), the foldase circuit. Foldases resemble enzymes inasmuch as they are saturable, so their processing rates can be modelled with Michaelis-Menten kinetics [15]. Fig 2 shows the Michaelis-Menten function, but turned sideways, to make a useful point; namely, if the influx rate from either protein synthesis or unfolding is too fast, it saturates the foldases and the level of unfolded molecules will sharply rise. The point of showing this is to indicate how large fluxes of unfolded proteins requiring foldases drive a need for holdases to protect those proteins while they wait. This point is useful later when we consider the effects of growth rate. Eq 2 shows how we compute the conversion rate $k_{U\rightarrow N}$ of unfolded to native protein in terms of $[U_{\text{free}}]$, binding affinities, and known rate constants. When the proteostasis system is

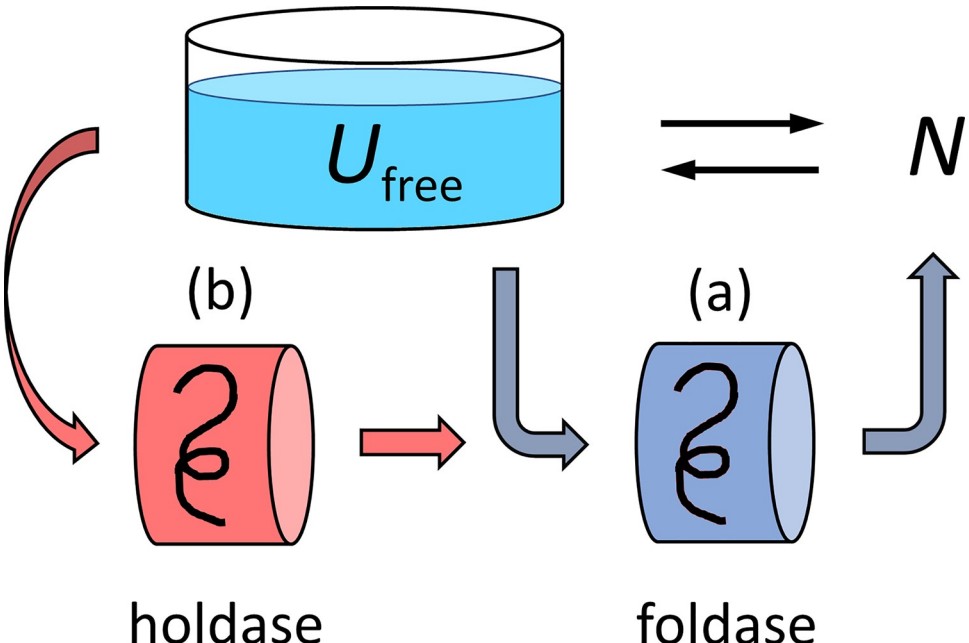

**Fig 1. Chaperone holdases and foldases traffic unfolded client proteins ($U_{\text{free}}$) to their native conformations ($N$).** Unfolded proteins can only exist in solution up to a level where they "overflow" into aggregation. Unfolded proteins can either bind directly to (a) a foldase system [13] or (b) a holdase that acts as a parking spot until it is taken up by a foldase [14].

in a steady state, i.e. when the concentration of unfolded protein $[U_{\text{free}}]$ is unchanging, the rates of flow out of $U_{\text{free}}$ must equal the rates of flow into $U_{\text{free}}$. This steady state condition allows us to express $[U_{\text{free}}]$ as a function of foldase level, synthesis rate and the folding stability of the client protein (Eq 3 and the curve in Figs 2 (right) and S1A).

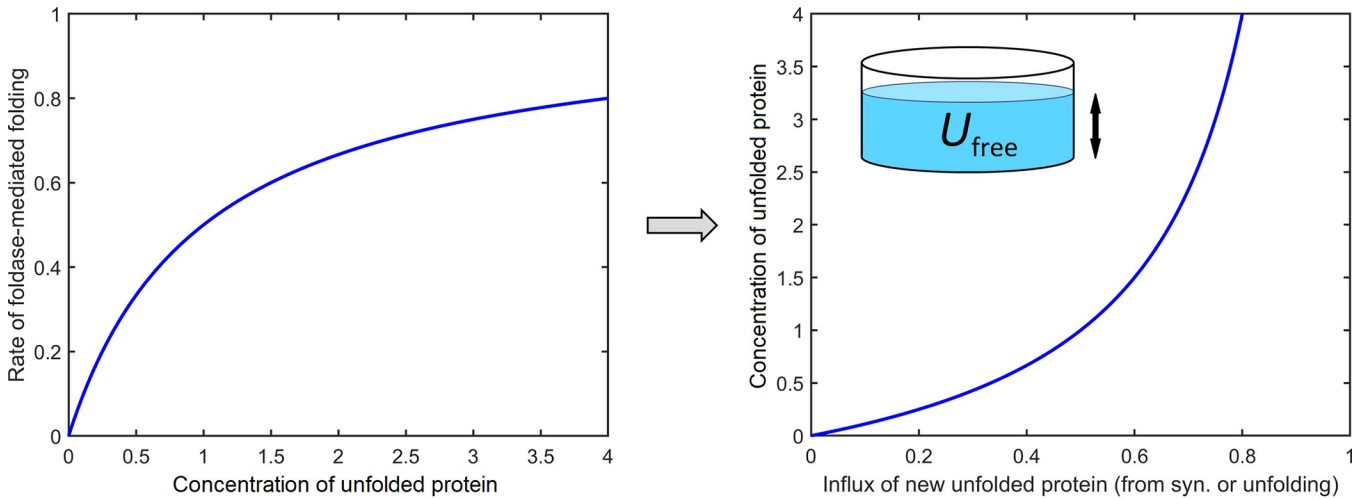

**Fig 2. Foldases follow Michaelis-Menten kinetics.** (Left) If client protein becomes too abundant, it saturates the foldases and unfolded protein accumulates. (Right) Reversing the axes shows how high input rates of unfolded protein can overflow the chaperone, leading to a sharp increase in unprocessed protein. For this situation, to keep these accumulating proteins from becoming lost to processes like aggregation, cells need holdases as "holding tanks" where client proteins can wait in relative safety for foldases to arrive.

Next, consider (b), the holdase circuit. Because of the separation of time scales between circuits (a) and (b), our calculations for the holdase can be made based on the fixed value of [$U_{free}$] established by the foldases. Eqs 5 and 6 show how excess unfolded client protein gets taken up by the "holding tank" of holdases, as a temporary repository, while waiting for a foldase to arrive. We assume that foldases bind holdase-bound protein at the same rates as they do non-bound $U_{free}$. The equilibrium concentration [$U_{free}$] in the presence of holdases can be expressed mathematically and solved in limiting cases, as shown in S1B Fig. Under low unfolding loads–where unfolding inflows are small compared to foldase capacity for outflows (eg: smaller box in S1A Fig)–holdases outnumber unfolded protein and are predicted to strongly deplete [$U_{free}$] (bottom-left corner of S1B Fig). By providing a safe, soluble route to foldases, holdases allow for greater foldase saturation–and hence specific activity (larger box in S1A Fig)–without excessive levels of [$U_{free}$]. In contrast, under very high unfolding loads, the pool of unfolded protein can exceed the carrying capacity even of the holdases, which become fully saturated and provide no further protection against unfolding (righthand side of S1B Fig), heightening the risk of functional loss through protein aggregation, oxidation or degradation [16–18]. The expression of holdases and foldases must be carefully tuned so as not to allow the burden on either chaperone system to become excessive, with their combined effect on [$U_{free}$] summarized by Eq 7.

## Calculating the costs of holdases *vs*. foldases

Now, we explore the idea that evolution has arrived at a principle of optimization–like a fitness function–that dictates the relative levels of expression of holdases and foldases. Given that energy is precious to the cell, we hypothesize that the two types of chaperones are expressed in a ratio that maximizes foldedness of client proteins at a minimum energy cost. Chaperone costs can be divided into an upfront synthesis cost proportional to the size of the chaperone and a running cost equal to the amount of energy (ATP) needed for function. For holdases, which do not consume ATP, the cost is just $c_{holdase} = c_{syn,holdase}$, where $c_{syn,holdase}$ is the cost to synthesize the holdase. For foldases, the cost can be expressed as $c_{foldase} = c_{syn,foldase} + c_{run,foldase}t$ (see S2 and S3 Figs), $c_{run,foldase}$ is the cost to run the foldase per unit time and $t$ is the duration it runs prior to its degradation or dilution ($\approx 1/k_{syn}$). This run rate can either be treated as a constant, independent of unfolding loads, or be divided into load-dependent and load-independent (background) activity. We choose to treat this run rate as a constant and explore the sensitivity to this assumption (see S1 Text and S4 Fig). For synthesis, we assume that adding one amino acid to a growing protein chain costs 5 ATP, equal to the cost of polymerization and proofreading [19] (the costs of the amino acids themselves can be included, increasing the overall synthesis costs, but this does not change the key conclusions). We propose that the running cost is a key driver of chaperone adaptation, especially in the limit of slow growth where running costs dominate.

## The cost-benefit trade-off reaches a peak when the marginal benefits are equal, of holdases *vs*. foldases

We explore the principal that the optimal ratio of holdases to foldases occurs when the marginal decrease in [$U_{free}$] from expressing more holdases is equal to the marginal decrease from expressing more foldases of equal cost. Eqs 8–13 state this principle in mathematical terms. The optimal ratio can be converted into chaperone abundances if we assume that cells regulate chaperone levels such that the risk of functional loss of a protein client over its lifetime is constant (sensitivity to this assumption is shown in S5 Fig). Given that proteins are more likely to become aggregated, damaged, or degraded when in non-native conformations [16–18], constant risk is assumed to be equivalent to holding constant the total time a protein molecule

spends in $U_{\text{free}}$ over its lifetime. As even native proteins unfold regularly–over hours or days depending on stability–a two-fold increase in a protein's lifetime needs to be accompanied by a two-fold reduction in the risk of functional loss *per unfolding event*. This reduction can be accomplished by faster detection and binding of vulnerable, unfolded protein by both holdases and foldases. We propose that this need for increased protein protection is a driver of chaperone adaptation in the limit of slow growth.

## The holdase-foldase ratio should increase at slow growth rates

Fig 3 shows the predictions of the energy optimization principle, obtained by applying Eqs 8–13 (see Methods) along with the condition that a protein's total unfolding dwell time is constant over its effective half-life. This half-life is determined by a combination of degradation and dilution, and thus is proportional to $1/k_{\text{syn}}$, where $k_{\text{syn}}$ is a measure of the proteome's overall synthesis rate. For simplicity, we will study the behavior of an average protein with turnover comparable to the mean of the proteome and use the same value for the turnover of our chaperones. An exploration of model sensitivity to assumptions can be found in S1 Text and S3–S6 Figs. Fig 3 shows the predicted optimal levels of the two chaperone types as a function of synthesis rate (a surrogate for the cell's growth rate in rapidly growing cells). The diagram is constructed as follows, from bottom to top. First, we show the optimal foldase level in blue. It counteracts two sources of unfolding: (i) due to the stochastic unfolding $N \rightarrow U$ of folded protein, which happens at a constant rate independent of protein synthesis, and (ii) due to protein synthesis, which is linear in the synthesis rate. Hence, the optimal foldase curve (blue) is flat at

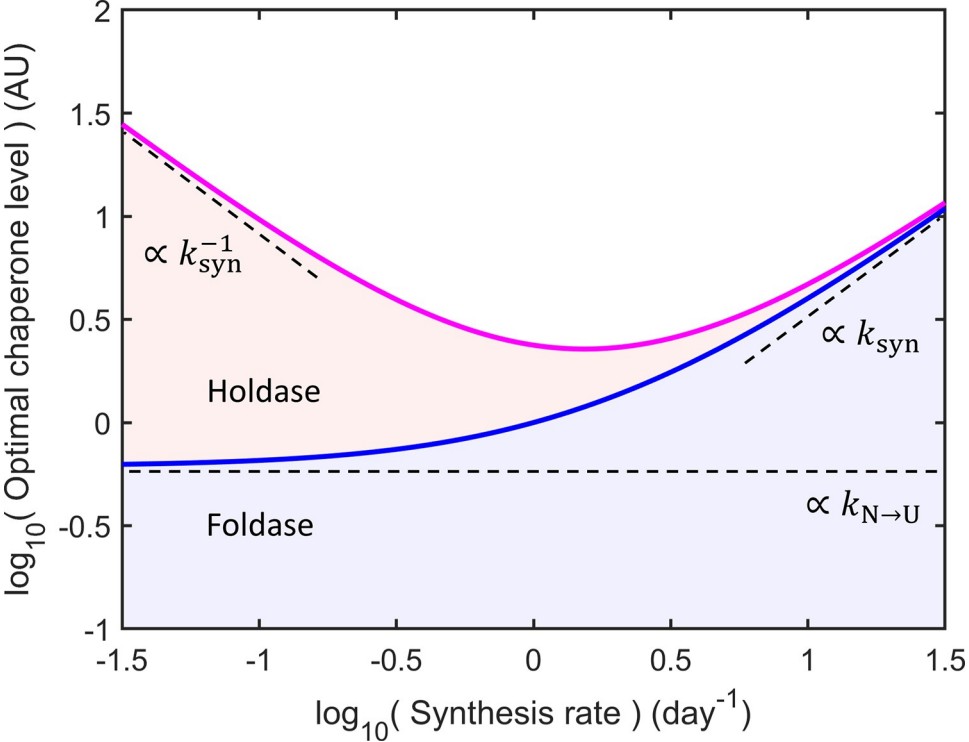

**Fig 3. Optimal ratio of holdases to foldases depends on synthesis rate of a cell.** The optimal allocation of foldases (blue) and foldases + holdases (magenta) to achieve fixed unfolded dwell time is predicted to vary with synthesis rate, $k_{\text{syn}}$. At low and high synthesis rates, unfolding loads are dominated by unfolding (proportional to $k_{\text{N}\rightarrow\text{U}}$) and synthesis (proportional to $k_{\text{syn}}$), respectively (see S3–S6 Figs). Holdase levels are predicted to rise inversely with $k_{\text{syn}}$.

the left and rises linearly at the right. Holdases, in contrast, are predicted to be most beneficial under slow growth. In the limit of slow synthesis rates (left), Eq 13 predicts that holdase levels should grow larger, scaling as $1/k_{syn}$ in proportion to effective half-lives. Holdase expression under fast synthesis is also predicted to rise, but with a slope that depends on model assumptions (the magenta curve in Fig 3 is holdase + foldase levels; see S3–S6 Figs for holdase levels alone).

Here is the essence of Fig 3. Take the protein synthesis rate as the independent variable. It dictates the replacement rate that is needed (for non-secretory cells). Now, to maintain a fixed level of proteome folding, holdases and foldases together need to hold constant a client molecule's total unfolded dwell time–its risk of loss. Lower replacement rates with new protein–and thus longer half-lives–necessitates lower levels of *free* (non-bound) unfolded protein at any given time simply because there is more time, and hence more unfolding events, over which it must remain safe. Yet, keeping the proteome folded using foldases alone is not cost-effective because it requires too high a concentration of ATP-consuming foldases (see Eq 2 and S1 Table). Holdases provide an affordable solution by holding the client and keeping it safe until a foldase arrives. While protein solubility and safety put a limit on the level of free, unfolded protein that can be tolerated, holdases provide a safe, *additional* route to foldases. This raises the activity per foldase molecule, lessening the need for their expression. So, creating the optimal chaperone allocation is a delicate balance depending on folding requirements and energy cost.

Fig 3 shows that under fast-synthesis conditions, the most cost-effective solution is to produce proportionately more foldases, as (i) the impact of their running costs is lessened by the shorter time window of operation ($1/k_{syn}$), and (ii) the shorter replacement time permits higher rates of client protein loss *per unit time*, and thus higher unfolded protein levels that allow greater flow through the foldase systems. These factors make foldases more cost-effective under fast-synthesis conditions.

The situation is different under slow-synthesis conditions. Here, the cell must keep its proteome safe over long times and thus more unfolding events. Keeping a protein safe for twice as long requires halving the threat of protein functional loss per unit time. This may be a core reason why organisms upregulate stress response machinery at slow growth *despite* the lower synthesis burden [20,21]. Over these longer timespans, ATP running costs begin to dominate the chaperone budget, making holdases proportionately better value (S2 Fig and Eq 13). So, under slow-synthesis conditions, holdases are more energy efficient.

Between these two growth extremes is a "Goldilocks zone". At this point, the burden of folding newly synthesized proteins is balanced by the burden of keeping proteins safe until their "replacement part" is made. This is the point where the proteome can be maximally careless because replacement parts come quickly, but not so quickly that it significantly burdens the chaperone systems.

## Aging is a form of stress that calls for more holdases

Aging is associated with a decline in productive protein synthesis across a broad range of organisms. This is particularly evident in the nematode worm *C. elegans*, which experiences a 15- to 30-fold reduction in protein synthesis rate during adult aging [4]. This decrease is accompanied by changes in chaperone expression (S2 Table), coordinated in part by daf-16 [9,22] which, interestingly, also plays a role in slowing synthesis [23]. Comparing the model with the known holdase and foldase systems [24], we find the changes in chaperone abundance to be consistent with the predictions from cost-benefit analysis, as shown in Fig 4A and 4B (S2 Table). While HSP60, HSP70 and HSP90 foldase levels change little, holdases of various kinds increase markedly and in rough proportion to the decrease in synthesis rate, as predicted,

albeit with variation among types. For example, the abundant holdase sip-1, found in muscle and germline, increases even more sharply than predicted. This may be due to changes in the relative volume of germline to soma with age or a shift in pH [25], which can destabilize proteins [26]. In contrast, the holdase hsp-17 shows almost no age-dependence, although with its low expression, it could work in parallel with other holdases. Interestingly, daf-16 activation slows protein turnover [27], suggesting it may have a more global role in orchestrating an economical preservation of the protein made during youth, many of which last as long as the worm.

This chaperone adaptation through life can be summarized by superimposing the lifetime trajectory of synthesis rates onto Fig 3, as shown in Fig 4C. While data in fast-growing larvae are sparse, gene expression immediately after larval development has been shown to correlate with the heat shock response, suggesting chaperone upregulation [28]. Further support comes from *C. elegans* insulin signalling mutants, which not only show a similar pattern of increased holdases and mildly decreased foldases with age [4], but also start adulthood with higher levels of holdases, consistent with their reduced synthesis rates when young [23,27]. These observations are not unique to worms. The ratio of holdases to foldases also increases in the aging human brain [24], for which there is evidence of slowed protein synthesis [29], consistent with an energy-preserving adaptation.

## Cost-efficiency predicts chaperone expression trends across species

If the present cost-benefit principle is general, it should apply not only *within* a single organism upon aging, but also *across* organisms in youth. To test this, we examined chaperone levels across a range of mammals. Holdase HSP25 levels are known across rodents [30] (Fig 5A), whereas several foldases have been measured across an even greater array of mammals [31] (Fig 5B). As protein synthesis rates are not known for all these organisms, we use metabolic rate as a proxy, as protein synthesis is a dominant component of metabolism and thus should scale similarly [1,32]. Observing the trends, we see that holdases follow an inverse relationship with metabolic rate, as predicted (red line in Fig 5A, slope = -1.00, $R^2$ = 0.61 excluding the outlier; slope = -1.33 with the outlier). In contrast to model predictions, foldase levels appear to *increase* with decreasing metabolic rate, albeit less than holdases and strongly dependent on the outlier (slope = -0.74 excluding the outlier; slope = +0.72 with the outlier).

To help resolve the uncertainty in the foldase dependence, we extended our comparison to other foldase types across a broader range of mammals [31]. Fig 5B shows the key findings. First, the average scaling exponent of -0.35 across four foldases–HSP60, HSP70, GRP78, and GRP94 –is much shallower than in Fig 5A, thus closer to model predictions. Second, the foldase with the greatest variation is HSP70 (blue), the same foldase type displayed in Fig 5A. This suggests that most foldases change less with metabolic rate than Fig 5A would imply, thus closer to the model prediction.

## Discussion

Here we have explored the hypothesis that organisms change the relative expression of holdases and foldases to maintain low levels of unfolded protein at the minimum energy cost. Applying this energy minimizing principle to a toy model of chaperone function predicts a U-shaped chaperone solution, where the most economical composition is foldases-heavy at fast growth, holdase-heavy at low growth (Fig 3), and requires the least chaperones in between. The counter-intuitive need for more chaperones at low synthesis rates, despite the lower

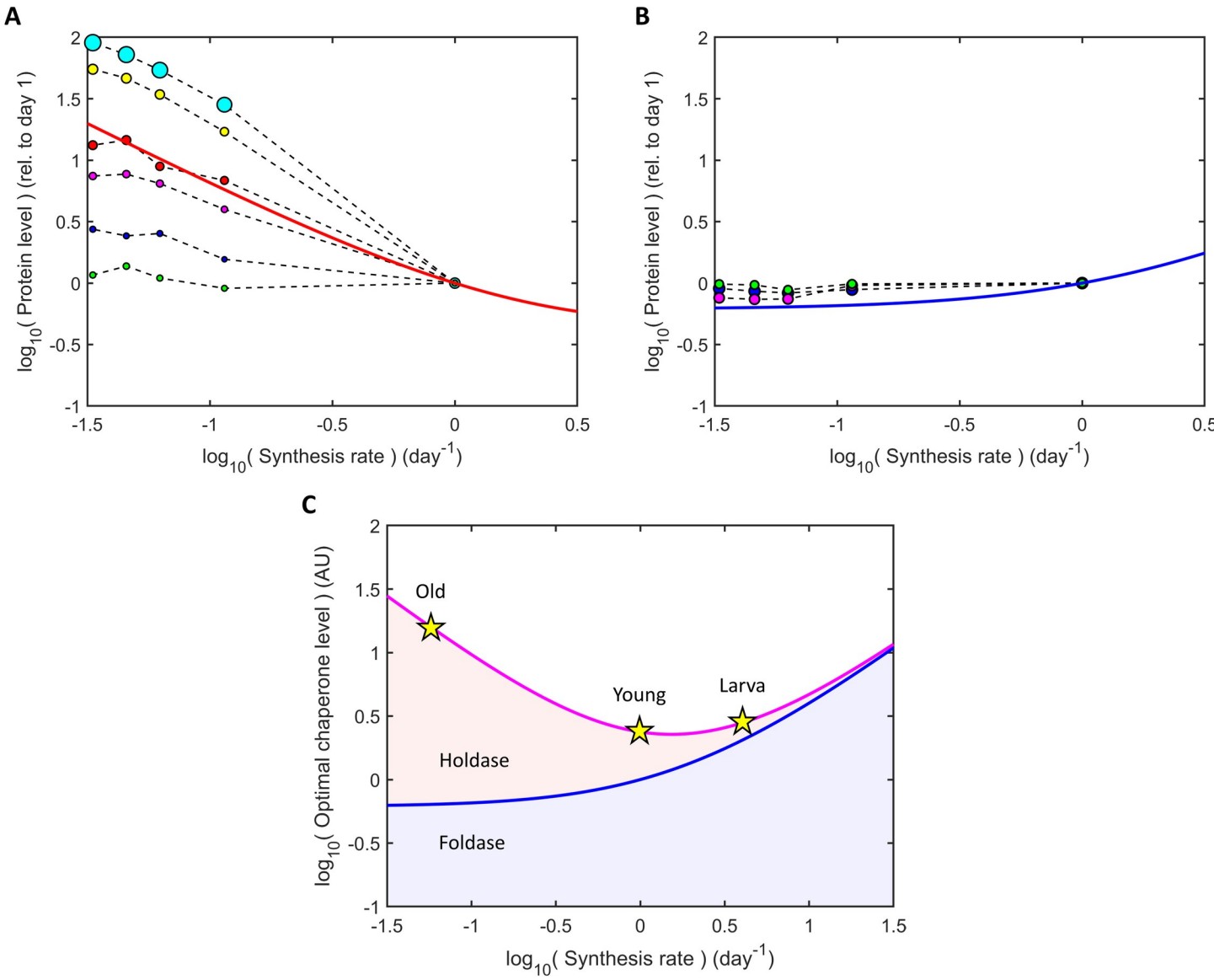

**Fig 4. Chaperone expression with age in *C. elegans* follows the predicted cost-efficient path.** (A) The expression of various holdases in *C. elegans* (top to bottom: sip-1, cian; hsp-43, yellow; hsp-16.48, red; hsp-12.3, magenta; Q9N350, blue; hsp-17, green) are compared to the predicted average behavior (red curve), where circle diameters reflect abundances, doubling for every 10-fold increase in concentration [4]. (B) Average abundance of *C. elegans* foldases (HSP60-class, green; HSP70-class, blue; HSP90-class, magenta; see S2 Table). The lowest two synthesis rates are extrapolated from their rates in middle age [4]. (C) *C. elegans* aging involves transitioning from a foldase-dominant to a holdase-dominant regime, estimated to cross a point of minimal chaperone need in young adulthood. Curves are the same as Fig 3. Stars show the predicted cost-efficient path.

burden of nascent protein, is supported by data from yeast [20] to *C. elegans* [4] to mammals [30,31]. It is even seen across the natural heterogeneity of cells within single tissues [34].

Despite this broad support for increased chaperone levels–and holdases in particular–at slow growth, the behavior of foldases is less uniform. While foldase levels are relatively constant in aging worms [4] and humans [24], they are modestly higher in mammals with lower metabolic rates. Why is this? A possible answer suggested by the model lies in the metabolic consequences of excess foldases. The model assumes that excess foldases result in an increase

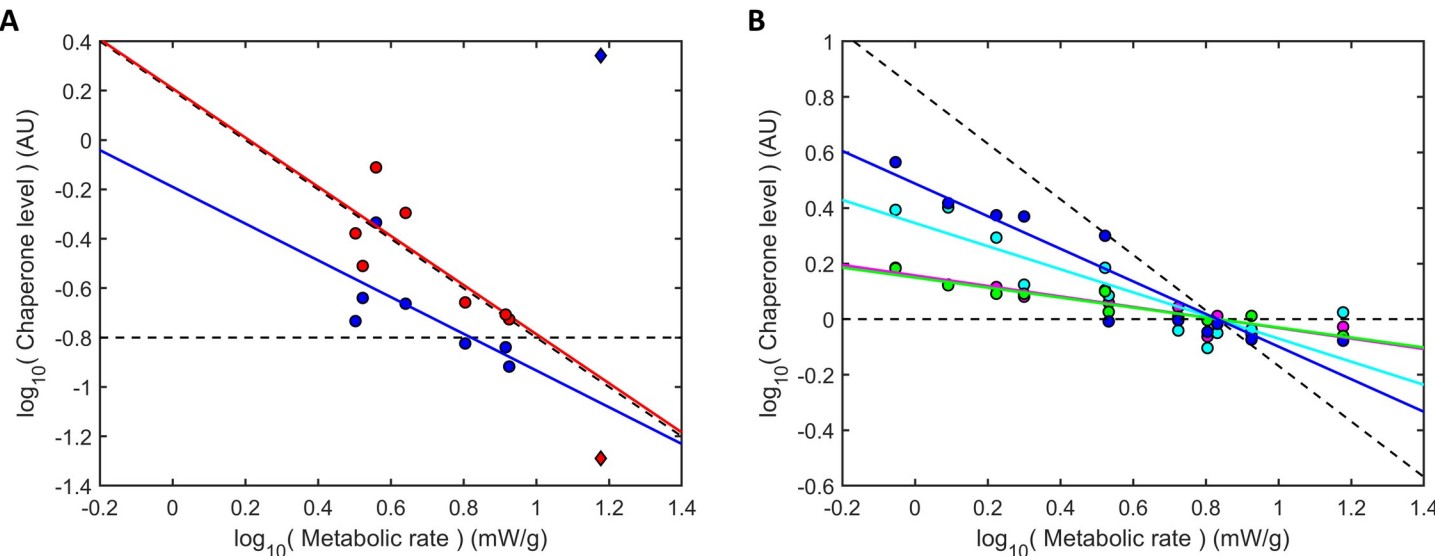

**Fig 5. Cost-efficient chaperone allocation is observed across young organisms.** (A) The scaling of holdase HSP25 (red; slope = -1.00, $R^2$ = 0.61) and foldase HSC70 +HSP70 (blue; slope = -0.74, $R^2$ = 0.50) levels in the quadriceps muscle of rodents [30] with basal metabolic rate [33] (mouse outlier in diamond excluded from fit). Dotted lines show the scaling slopes of -1 and 0 predicted from cost-efficiency at slow growth (Fig 2 and Eq 13). (B) Levels of foldase HSP60 (green; slope = -0.18, $R^2$ = 0.90), HSP70 (blue; slope = -0.59, $R^2$ = 0.86), GRP78 (magenta; slope = -0.19, $R^2$ = 0.82), and GRP94 (cian; slope = -0.42, $R^2$ = 0.77) across a broader set of mammals in liver [31]. Similar trends were found in brain and heart. Dotted lines are the same as (A).

in basal energy consumption. Such consumption could result from over-binding of unfolded protein, and thus to less efficient folding, or from excess foldases promoting greater cellular ATP consumption by indirect means [35]. In such scenarios, the model predicts foldase levels should only depend weakly on synthesis rates within the slow-synthesis regime expected for adult mammals (see Figs 3 and 4C and S4A). However, if excess foldases result in negligible additional energy consumption relative to their synthesis costs–the "efficient chaperone buffer" limit–then a different chaperone expression pattern is predicted to be optimal, with both foldases and holdases approaching a common slope of -0.5 (see S4B Fig). In other words, the additional need for chaperones at low metabolic rates are split equally between foldases and holdases. This is close to the average slope of -0.35 observed here across mammals [30,31], suggesting that excess foldases result in little additional ATP consumption when overexpressed. In contrast, excess foldases in an aging cell filled with irreparably altered, destabilized proteins may lead to high ATP consumption, a cost that would be hard for the cell to bear. Future experiments will be needed to test these hypotheses.

Much of biology are stories about the adaptation of cells to their environment. Protein folding accounts for a major component of cell machinery in terms of biomass, energy usage and complexity. It, too, must adapt when cells see different environments or growth conditions. What are the decision variables and actions the cell takes to make proteostasis adaptive? We have explored here a biophysical principle, namely that the cell adjusts its balance of chaperone holdases versus foldases to minimize its energy cost while maintaining a fixed average unfolded dwell time over the proteome. This principle predicts a U-shaped chaperone expression pattern, with greater need for chaperones at slow and fast growth than at medium rates of synthesis. At fast growth, foldases are needed in greater proportion to holdases, while at slow growth and in older cells, relatively more holdases are needed. Existing evidence is consistent with these predictions, providing a framework for interpreting chaperone expression across species, growth conditions and aging.

## Methods

### Analytical solution to the energy minimization problem

The most cost-effective combination of holdases and foldases can be solved numerically, as was performed for all figures in this work. However, insight can be gained by finding the analytical solution in limiting cases, as follows.

In the presence of foldases, the rate of change of unfolded protein concentration is the difference between the rate of creation and the rate of removal, given by

$$\frac{d}{dt}([U]) = (v_{syn} + v_{N \to U}) - \left(v_{foldase}\right) \tag{1}$$

$$\text{where } v_{foldase} = \frac{k_{foldase}[foldase]}{1 + \frac{K_{m,foldase}}{[U]}} \tag{2}$$

using Michaelis-Menten kinetics for $v_{foldase}$. The flux of newly synthesized foldase clients $v_{syn} = k_{syn}[clients]$ and newly unfolded foldase clients $v_{N \to U} = k_{N \to U}[clients]$ can be written in terms of the total concentration of foldase-dependent protein, $[client]$. Synthesis flux is not to be confused with elongation rate, which is distinct and not treated here. In steady state $\frac{d}{dt}([U]) = 0$, from which Eq 1 becomes

$$\frac{[U]}{K_{m,foldase}} = \frac{v_{syn} + v_{N \to U}}{k_{foldase}[foldase] - (v_{syn} + v_{N \to U})} = \frac{1}{\frac{k_{foldase}[foldase]}{v_{syn} + v_{N \to U}} - 1} \tag{3}$$

which is plotted in Figs 1B and S1A. In the limit that $k_{foldase}[foldase] \gg (v_{syn} + v_{N \to U})$ far from saturation, the system is in the linear region at the bottom left of S1A Fig, which can be approximated as

$$\frac{[U]}{K_m} \approx \frac{v_{syn} + v_{N \to U}}{k_{foldase}[foldase]} \tag{4}$$

Now, *given* the equilibrium in Eq 4, we can introduce holdases into the mix. If holdases are taken to act as passive, reversibly binding "holding tanks" that soak up unfolded protein [14] with a dissociation constant $K_d$ [6,36], the binding-unbinding equilibrium can be expressed as

$$K_d = \frac{[U][holdase]}{[U : holdase]} \tag{5}$$

where $[holdase]$ is the concentration of the multimeric functional unit, not necessarily the monomer [37]. Eq 5 can be solved in three limiting cases: (i) $[holdase_{tot}] \ll [U_{tot}] \& K_d$, (ii) $K_d \ll [holdase_{tot}] \& [U_{tot}]$, and (iii) $[U_{tot}] \ll [holdase_{tot}]$. Here, $[holdase_{tot}] = [holdase] + [U : holdase]$ and $[U_{tot}] = [U] + [U : holdase]$. In the low-holdase limit of case (i), holdases are inconsequential to proteostasis due to their low abundance, leading to

$$[U] \approx [U_{tot}] \quad (\text{for } [holdase_{tot}] \ll [U_{tot}] \& K_d) \tag{6A}$$

In the tight-binding limit of case (ii), holdases are fully bound, giving

$$[U] \approx [U_{tot}] - [holdase_{tot}] \quad (\text{for } K_d \ll [holdase_{tot}] \& [U_{tot}]) \tag{6B}$$

Lastly in the low-unfoldedness limit of case (iii), holdases are significant but unsaturated, giving

$$[U] \approx \frac{[U_{tot}]}{2} \cdot \left( 1 + \frac{K_d - [holdase_{tot}]}{K_d + [holdase_{tot}]} \right) \quad (\text{for} [U_{tot}] \ll [holdase_{tot}]) \tag{6C}$$

which further simplifies to

$$[U] \approx [U_{tot}] \cdot \frac{K_d}{[holdase_{tot}]} \quad (\text{for} [U_{tot}] \& K_d \ll [holdase_{tot}]) \tag{6D}$$

The limiting behaviours of Eqs 6B and 6C are shown in S1B Fig, for which case 6D is the most physiologically relevant, as holdases are believed to be mostly free of clients under normal conditions and outnumber the load of unfolded protein ($[U_{tot}] < 2$ μM, $K_d \approx 2$ μM [37], and $[holdase_{tot}] = 3$–$30$ μM) [38] (S1 Table). Given that foldases determine the total pool size of unfolded protein (Eq 4), whereas holdases determine the fraction of this pool that is non-bound and vulnerable (Eq 6C), these two factors are multiplicative. Solving Eq 4 for $[U]$ and multiplying by the fraction of $[U]$ that is free, namely $[U]/[U_{\text{tot}}]$, from Eq 6C gives

$$[U] \approx \frac{K_m(v_{syn} + v_{N \to U})/k_{foldase}}{[foldase]} \cdot \frac{K_d}{[holdase]} \tag{7}$$

The optimal expression of foldases and holdases is reached when the marginal decrease in $[U]$ from expressing more foldases is equal to the marginal decrease in $[U]$ from spending the same cost on more holdases. Mathematically, this can be written as

$$\frac{\partial [U]}{\partial cost_{foldase}} = \frac{\partial [U]}{\partial cost_{holdase}} \tag{8A}$$

Applying chain rule gives

$$\frac{\partial [U]}{\partial [foldase]} \cdot \frac{\partial [foldase]}{\partial cost_{foldase}} = \frac{\partial [U]}{\partial [holdase]} \cdot \frac{\partial [holdase]}{\partial cost_{holdase}} \tag{8B}$$

As the marginal costs $\frac{\partial cost_{foldase}}{\partial [foldase]}$ and $\frac{\partial cost_{holdase}}{\partial [holdase]}$ are independent of the chaperones' expressions, they can be removed from the derivatives and Eq 8B becomes

$$\frac{\partial [foldase]}{\partial cost_{foldase}} \frac{\partial [U]}{\partial [foldase]} = \frac{\partial [holdase]}{\partial cost_{holdase}} \frac{\partial [U]}{\partial [holdase]} \tag{8C}$$

Given that $\frac{\partial cost_{foldase}}{\partial [foldase]}$ and $\frac{\partial cost_{holdase}}{\partial [holdase]}$ are the unit costs, namely $c_{foldase} = c_{\text{syn,foldase}} + c_{\text{run,foldase}} t$ and $c_{holdase} = c_{\text{syn,holdase}}$, Eq 8C becomes

$$\frac{1}{c_{foldase}} \frac{\partial [U]}{\partial [foldase]} = \frac{1}{c_{holdase}} \frac{\partial [U]}{\partial [holdase]} \tag{9}$$

Taking the derivatives of Eq 7, $\frac{\partial [U]}{\partial [foldase]} = -\frac{[U]}{[foldase]}$ and $\frac{\partial [U]}{\partial [holdase]} = -\frac{[U]}{[holdase]}$, Eq 9 becomes

$$\frac{1}{c_{foldase}} \frac{[U]}{[foldase]} = \frac{1}{c_{holdase}} \frac{[U]}{[holdase]} \tag{10}$$

Finally, inserting the unit costs gives the optimal chaperone ratio that minimizes unfoldedness $[U]$,

$$\frac{[holdase]}{[foldase]} = \frac{c_{foldase}}{c_{holdase}} = \frac{c_{\text{syn,foldase}} + c_{\text{run,foldase}}t}{c_{\text{syn,holdase}}} \tag{11}$$

The foldase run time is its effective half-life from degradation and dilution, ie. $t = \tau_{1/2} \, \alpha \, 1/k_{syn}$, giving

$$\frac{[holdase]}{[foldase]} = a + b\frac{1}{k_{\text{syn}}} \tag{12}$$

where $a = \frac{c_{\text{syn,foldase}}}{c_{\text{syn,holdase}}}$ and $b = \frac{c_{\text{run,foldase}}}{c_{\text{syn,holdase}}}$

In the limit of slow synthesis ($k_{syn}$ is small and $\tau_{1/2}$ is large), foldase running cost dominates, thus

$$\frac{[holdase]}{[foldase]} \approx b\frac{1}{k_{\text{syn}}} \tag{13}$$

Assumptions used for this derivation fail at fast synthesis, where low foldase usage (linear approximation of Michaelis-Menten kinetics) and low unfolding levels no longer hold. For a solution over all synthesis rates, numerical optimization was performed, with the results shown in Figs 3 and 4.

## Supporting information

**S1 Fig. Foldases and holdases each decrease the level of unfolded protein.** (A) Foldases actively promote the folding of proteins that are newly unfolded (flux $v_{\text{N}\rightarrow\text{U}}$) or newly synthesized ($v_{syn}$). The total foldase capacity ($v_{foldase}$) needs to be greater than the unfolding burden ($v_{\text{N}\rightarrow\text{U}} + v_{syn}$) to prevent unfolded levels from diverging (vertical dashed line is the limit of the Michaelis-Menten function describing foldase activity in blue). The benefit of a large spare foldase capacity can be seen from the rapid rise of the unfolded level $[U]$, up 4-fold for only a 2-fold increase in unfolding load (thin black lines). (B) Holdases further deplete the level of unfolded protein by acting as protective "parking spots" while their unfolded clients wait for a foldase. The degree of depletion, given by the distance of the curve below the diagonal, can be solved in limiting cases (thin dashed lines, see Methods for the derivation).
(TIF)

**S2 Fig. Chaperone costs depend on usage time.** Total foldase cost (blue line) is the sum of an initial synthesis cost and an increasing "running cost" (shaded area) whereas holdases do not consume ATP and have a flat synthesis cost (red line). The running cost of an HSP70-type foldase in constant use is estimated to equal the synthesis (polymerization) cost after roughly 10 hours (dashed line). This stretches to roughly 100 hours for HSP60-class chaperones due to cycle times that are an order of magnitude slower (ATP consumption per kDa per cycle is comparable to HSP70 chaperones) (see S4 Fig).
(TIF)

**S3 Fig. Sensitivity to foldase size or synthesis cost.** Effect on optimal chaperone expression if foldase size or synthesis cost is increased 10-fold from 70 kDa (3,500 ATP) to 700 kDa (35,000 ATP).
(TIF)

**S4 Fig. Sensitivity to foldase activity.** (A) Effect on optimal chaperone expression if foldase activity goes from "always on" to 10% ATPase activity when not folding proteins (ATP-dependent chaperones have functions other than folding). (B) In the limit of "perfect foldases" that have zero ATPase activity when not folding proteins, foldases "split the difference" with holdases, each scaling as $1/\sqrt{k_\text{syn}}$.
(TIF)

**S5 Fig. Sensitivity to unfoldedness requirements.** Effect on optimal chaperone expression if unfolding requirement goes from $[U_\text{free}] = u_\text{o}k_\text{syn}$ to $[U_\text{free}] = 100$ nM, where $u_\text{o} = 100$ nM·day (see S1 Table).
(TIF)

**S6 Fig. Sensitivity to holdase binding affinity for non-native proteins.** Effect on optimal chaperone expression from holdases transitioning from a loose binding ($K_\text{d} = 1$ μM; light curves) to tight binding regime ($K_\text{d} = 10$ nM; dark curves). Holdases (red) and foldases (blue).
(TIF)

**S1 Text. Dependence of optimal chaperone expression on chaperone properties.**
(DOCX)

**S1 Table. Parameters of the model.**
(DOCX)

**S2 Table. Experimental measurements of *C. elegans* chaperone levels.**
(XLSX)

## Acknowledgments

We are grateful to Ursula Jakob, Johannes Buchner and Mantu Santra for fun and insightful discussions. We appreciate the support from the Stony Brook University Laufer Center.

## Author Contributions

**Conceptualization:** Adam MR de Graff, Ken A. Dill.

**Data curation:** Adam MR de Graff.

**Formal analysis:** Adam MR de Graff, David E. Mosedale.

**Software:** Adam MR de Graff.

**Supervision:** David E. Mosedale, Tilly Sharp, Ken A. Dill, David J. Grainger.

**Visualization:** Adam MR de Graff, David E. Mosedale, Tilly Sharp, Ken A. Dill, David J. Grainger.

**Writing – original draft:** Adam MR de Graff, David E. Mosedale, Tilly Sharp, Ken A. Dill, David J. Grainger.

**Writing – review & editing:** Adam MR de Graff, David E. Mosedale, Tilly Sharp, Ken A. Dill, David J. Grainger.

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
