## [Decision Letter · Decision Letter 0]

9 Sep 2020

Dear de Graff,

Thank you very much for submitting your manuscript "Proteostasis is adaptive: balancing chaperone holdases against foldases" for consideration at PLOS Computational Biology. As with all papers reviewed by the journal, your manuscript was reviewed by members of the editorial board and by several independent reviewers. The reviewers appreciated the attention to an important topic. Based on the reviews, we are likely to accept this manuscript for publication, providing that you modify the manuscript according to the review recommendations.

Based on our reading of the paper and the reviewers' recommendations, this paper is recommended with minor revision to address the technical comments of the third reviewer. We look forward to receiving a revised version in the near future.

Sincerely,

Jeffrey Skolnick

Guest Editor

PLOS Computational Biology

Nir Ben-Tal

Deputy Editor

PLOS Computational Biology

[LINK]

Based on my reading of the paper and the reviewers' recommendations, this paper is recommended with minor revision to address the technical comments of the third reviewer. I look forward to receiving a revised version in the near future.

Reviewer's Responses to Questions

**Comments to the Authors:**

Reviewer #1: This is a very nice study, which can be published as is.

Reviewer #2: Reproducibility Report has been uploaded as an attachment.

Reviewer #3: This manuscript is concerned with how the cell balances the folded state of the proteome through expression of chaperones which either accelerate folding (foldases) or stablise the protein whilst awaiting folding (holdases). The authors present a brief, yet elegant, account of how the holdase:foldase ratio might be changed to provide the optimal proteome folding efficiency in terms of ATP consumption.

Whilst a purely theoretical study, the authors show that their model convincingly describes existing experimental data from C. elegans and various rodents, applying the model to both the same organism at different ages, as well as across different organisms.

Pending some minor changes, I would recommend publication of the manuscript:

• Near the bottom of page 5: ‘raise’ should be ‘raises’

• ‘Walther 2015’ in the text needs to be formatted to match the other citations

• I’m a little confused by the red and blue lines in Figs 4a and b. Are they simply the model predictions from Fig 3 shifted on the y-axis such that day 1 = 0? I feel this could be made clearer in the text.

• Perhaps some remarks could be made about the completeness of the data in Figs 4a and b. Are all known holdases/foldases included? If not, why were the selected examples chosen.

• Can anything be stated about the substantial range in the data used for Fig 4a, i.e. why is hsp-17 level seemingly independent of Ksyn?

• It is stated in the text that the foldase least-squares fit in 6A is “highly dependent on the outlier”, yet it seems from the figure legend that the outlier has actually been excluded from the fit. This should be checked and clarified. Also, the outlier should be marked somehow on the graph (e.g. different symbol)

• The goodness of fit (R^2 or similar) of the least-squares fits used in Figure 6 should be reported.

• It seems that the foldase data in Fig 6A might better be described by the “perfect foldase” model in Fig S4. Could it be that foldase efficiency is different in younger organisms, explaining the disparity between the model and experimental data for foldase levels vs metabolic rate in 6A?

• The title of ref 1 appears to be truncated

**Have all data underlying the figures and results presented in the manuscript been provided?**

Reviewer #1: Yes

Reviewer #2: None

Reviewer #3: Yes

PLOS authors have the option to publish the peer review history of their article (what does this mean?). If published, this will include your full peer review and any attached files.

Reviewer #1: No

Reviewer #2: **Yes: **Anand K. Rampadarath

Reviewer #3: **Yes: **Robin Corey
---

## [Editor Report · Decision Letter 1]

20 Oct 2020

Dear de Graff,

We are pleased to inform you that your manuscript 'Proteostasis is adaptive: balancing chaperone holdases against foldases' has been provisionally accepted for publication in PLOS Computational Biology.

Best regards,

Jeffrey Skolnick

Guest Editor

PLOS Computational Biology

Nir Ben-Tal

Deputy Editor

PLOS Computational Biology

---

## [Editor Report · Acceptance letter]

26 Nov 2020

PCOMPBIOL-D-20-01214R1 

Proteostasis is adaptive: balancing chaperone holdases against foldases

Dear Dr de Graff,

I am pleased to inform you that your manuscript has been formally accepted for publication in PLOS Computational Biology. Your manuscript is now with our production department and you will be notified of the publication date in due course.

With kind regards,

Nicola Davies
